# Metamaterial Cell-Based Superstrate towards Bandwidth and Gain Enhancement of Quad-Band CPW-Fed Antenna for Wireless Applications

**DOI:** 10.3390/s20020457

**Published:** 2020-01-14

**Authors:** Samir Salem Al-Bawri, Md Shabiul Islam, Hin Yong Wong, Mohd Faizal Jamlos, Adam Narbudowicz, Muzammil Jusoh, Thennarasan Sabapathy, Mohammad Tariqul Islam

**Affiliations:** 1Faculty of Engineering, Multimedia University, Persiaran Multimedia, Cyberjaya 63100, Selangor, Malaysia; shabiul.islam@mmu.edu.my (M.S.I.); hywong@mmu.edu.my (H.Y.W.); 2Department of Electronics & Communication Engineering, Faculty of Engineering & Petroleum, Hadhramout University, Al-Mukalla 50512, Hadhramout, Yemen; 3Faculty of Mechanical Engineering, Universiti Malaysia Pahang, Pekan 26600, Pahang, Malaysia; 4Department of Telecommunications and Teleinformatics, Wroclaw University of Science and Technology, Wroclaw 50-370, Poland; 5Bioelectromagnetics Research Group (BioEM), School of Computer and Communication Engineering, Universiti Malaysia Perlis (UniMAP), Kampus Pauh Putra, Arau, Perlis 02600, Malaysia; 6Department of Electrical, Electronic and Systems Engineering, Faculty of Engineering and Built Environment, Universiti Kebangsaan Malaysia, UKM, Bangi 43600, Selangor, Malaysia

**Keywords:** coplanar waveguide (CPW) antenna, near-zero refractive index (NZRI), wideband, DNG metamaterial, multiband

## Abstract

A multiband coplanar waveguide (CPW)-fed antenna loaded with metamaterial unit cell for GSM900, WLAN, LTE-A, and 5G Wi-Fi applications is presented in this paper. The proposed metamaterial structure is a combination of various symmetric split-ring resonators (SSRR) and its characteristics were investigated for two major axes directions at (x and y-axis) wave propagation through the material. For x-axis wave propagation, it indicates a wide range of negative refractive index in the frequency span of 2–8.5 GHz. For y-axis wave propagation, it shows more than 2 GHz bandwidth of near-zero refractive index (NZRI) property. Two categories of the proposed metamaterial plane were applied to enhance the bandwidth and gain. The measured reflection coefficient (S11) demonstrated significant bandwidths increase at the upper bands by 4.92–6.49 GHz and 3.251–4.324 GHz, considered as a rise of 71.4% and 168%, respectively, against the proposed antenna without using metamaterial. Besides being high bandwidth achieving, the proposed antenna radiates bi-directionally with 95% as the maximum radiation efficiency. Moreover, the maximum measured gain reaches 6.74 dBi by a 92.57% improvement compared with the antenna without using metamaterial. The simulation and measurement results of the proposed antenna show good agreement.

## 1. Introduction

Recently, the development of metamaterial has received significant interest as a material for the creation of numerous novel structures with unconventional electromagnetic properties and noticeably enhanced performances [1,2]. Several metamaterials have been created with negative real parts for both permittivity (ɛ) and permeability (µ)—so-called double-negative metamaterials (DNG). This is in addition to the single-negative metamaterials (SNM) where either µ or ε is negative, called mu-negative metamaterial (MNG) and epsilon-negative metamaterial (ENG), respectively [3,4]. The first DNG was theoretically described by Veselago in 1968 [5] and experimentally verified later in [6]. The DNG supports an electromagnetic wave where the phase and group velocities propagate in opposite directions. This allows new electromagnetic properties [7,8] and applications in antenna design, biomedical, and wireless technologies [9]. However, near-zero refractive (NZRI), as well as epsilon-and-mu-near-zero (EMNZ) properties, have been investigated for many metamaterial structures in the state-of-the-art for specific applications such as S, C, X, and KU-bands [10,11,12,13]. Today, the demand for designing multiband antennas with a low cost and minimal weight has become highly essential due to the presence of multiple communication standards. Today, antenna performance has been significantly improved using metamaterial, which offers improved size miniaturization, gain, bandwidth, as well as decreased cost [14,15,16].

Several types of metamaterial structures have been proposed: a split-ring resonator (SRR) [17,18], complementary split-ring resonators (CSRRs) [19,20], periodic array structures [21], and spiral resonators (SR) [22]. For instance, SRR unit cell structures have been investigated for antenna design miniaturization, creating multiband operation, and improving bandwidth and gain [23,24]. Moreover, several multiband antennas loaded with metamaterial have been reported in [25,26,27,28], where there are trade-off characteristics, including the number of operating bands, size, and gain. However, miniaturization of antenna size can be attained using substrates with high permittivity or small size radiating elements [29]. The antenna radiator size can be decreased through various methods, including applying metamaterial, creating slots, using defected ground structure, and adding parasitic element strips.

In this paper, we propose a multiband CPW-fed antenna array covered with a planar investigated metamaterial (MTM) superstrate based on a near-zero refractive index (NZRI) and double-negative metamaterials (DNG) over a wide frequency range. The proposed MTM unit cell structure consists of several symmetric split-ring resonators (SSRRs) in order to exhibit real values of negative permeability and permittivity. The proposed antenna simultaneously operates within a GSM900 bandwidth at 0.9 GHz, WLAN at 2.3/2.4/5/5.2/5.8 GHz, 4th generation LTE-A at 3.5 GHz, and 5G Wi-Fi in the frequency range of 5.15–5.875, while it has an overall size of 78.6 × 42.5 mm^2^ (0.236λ_0_ × 0.128λ_0_ with respect to its lowest operating frequency). Simulated and measured results (with and without MTM) are in good agreement with the simulation. An overview of the different multiband CPW antennas published in the literature is presented in Table 1 for comparison with the proposed design. It can be seen that the proposed antenna offers significantly increased bandwidth and gain over similar antennas.

## 2. Metamaterial Unit Cell Design Architecture 

Figure 1a shows the front view of the proposed unit cell structure. It consists of several pairs of symmetric C-shaped split-ring resonators that are assembled and cover each other to enable the unit cell operation over the wideband range. The unit cell was designed on an FR4 substrate with 4.6 substrate dielectric constant, and a thickness of 1.6 mm. Here, two simulation set-ups were applied to verify the unit cell working principle. Firstly, the structure was located between two-wave guide ports in the x-direction as shown in Figure 1b, then in the y-direction as illustrated in Figure 1c. This was simulated using a finite-difference time-domain solver based on computer simulation technology (CST). 

The simulation of metamaterial was executed for the frequency range of 2–20 GHz. The negative electric permittivity and permeability are usually used as the material parameters that describe how materials polarize in the presence of electric and magnetic fields, which leads to achieving a negative refractive index.

## 3. Metamaterial Working Principle

To observe the proposed metamaterial behavior and understand the physical phenomena of how it works when it is located in an electric and magnetic field region, the surface current distributions are analyzed and discussed for different frequencies.

The surface current distributions of the proposed MTM unit cell at 2.4, 3.5, 5.5, and 10 GHz are shown in Figure 2a–d. The arrows represent the current distribution direction in the overall structure, and the colors indicate the current intensity. A visible surface current distribution at 2.4 GHz resonant frequency is illustrated in Figure 2a. However, more intensive surface current can be observed clearly at 3.5 GHz in Figure 2b, especially at the edge of outer symmetric C-shaped portions, whereas it perturbed the overall unit cell structure. In Figure 2c,d, the surface current distribution behavior is relatively in fluctuation mode at the symmetrical C-shaped outer, middle, and inner rings, whereas currents flow in different directions with a stronger current at 10 GHz. Due to this mechanism, an effective negative permeability is observed, although the current distribution flows in opposite side directions of the down and upper C-shaped etching strips, which nullify the current and generate a stopband.

The simulated transmission coefficient (S21) and reflection coefficient (S11) results of the x- and y-directions are shown in Figure 3a,b, respectively. For the x-direction, it shows the resonance frequency band in the range of 2–8.6 GHz, which is a part of UWB as well as covering several bands in the ranges of 9.8–10, 13.5–15, and 17–19.6 GHz. The outer and inner split-ring-shaped resonators are considered as the effective cause of the achieved multiband range, whereas the higher resonance at 10 GHz is shown by the blue dotted line. Furthermore, several bands are considered in the case of MTM located in the y-direction, where the S21 is in the range of 2–4.6, 5.5–6.5, 9.8–11.2, and 12–16.9 GHz.

The simulated relative imaginary and real parts of the effective permittivity, permeability, refractive index, and impedance of different unit cell array structures are plotted in Figure 4, Figure 5 and Figure 6. In these figures, the dark brown color is used to show the negative indexed zone for double-negative metamaterial (DNG) or single-negative metamaterial (SNG); however, light brown color indicates a near-zero refractive index (NZRI) and both Epsilon/Mu near-zero (ENZ/MNZ) metamaterial. The wide DNG real value of the refractive index is exhibited in the range of 2–8.5 GHz in x-axis wave propagation as shown in Figure 4a, whereas a narrow portion of NZRI is achieved with a 255 MHz bandwidth. However, it is remarkable that a greater than 2 GHz bandwidth can be achieved with an NZRI property in y-axis wave propagation, as shown in Figure 4b. This has potential applications for electromagnetic cloaking operation and high gain antenna design. For the 1 × 1 unit cell, as shown in Figure 5, real negative values of permittivity (ε) and permeability (μ) are exhibited for frequency ranges from 2.6 to 8.8 and from 2 to 6.4 GHz, respectively. Thus, the unit cell behaves as a double-negative metamaterial (DNG). Moreover, X band and KU bands are covered using the proposed metamaterial only in x-axis wave propagation with MNZ property.

In Figure 6, it is apparent that several bands are exhibited with negative real values of SNG permittivity, whereas SNG permeability in the range of 2–4.3 GHz is realized, which works for S-band applications and can cause NZRI in the same range. However, wide MNZ is clearly displayed at the real magnitude of effective permeability in the range of 7.2–13.8 GHz.

Figure 7 shows the simulated relative refractive index for the different MTM of the 1 × 1, 1 × 2, and 2 × 2 array structures. In the case of the x-direction, NZRI has been exhibited over a wide range of 10–20 GHz, using 1 × 2 or 2 × 2 MTM array, which includes the resonances in the X and KU bands compared with a non-index in the case of 1 × 1 MTM. However, the common effective NZRI bandwidth in y-axis wave propagation expands from 2.38 GHz in the case of the 1 × 1 and 1 × 2 MTM array to 3.1 GHz for the 2 × 2 MTM array structure.

## 4. Configuration of the Proposed Antenna

Figure 8a–d illustrates the simulated geometry of the proposed antenna. It can be seen as two compact elements, each forming a triangular radiator structure. The antenna is fed by a narrow 50 Ω CPW transmission line with a feedline length of *l_1_* = 8.86 mm, width *w_4_* = 2.44 mm, and two identical gaps of width *s* = 1 mm. It is followed by a T-junction divider which has two thin arms with a 0.42 width for each one, to correspond with the 100 Ω line. The ground and the radiator planes are printed on a low-cost, 1.6 mm thick FR4 substrate (with a relative permittivity of 4.6 and loss tangent of 0.025). The metamaterial is applied in two locations: One is in the unit cells placed at the backside of the antenna array and the other is suspended within a 10 mm distance away from the antenna where the unit cells are used. Several aspects were considered in order to optimize and miniaturize the proposed antenna and the novel unit cell using an Agilent advanced design system (ADS) and computer simulation technology (CST), respectively. The final geometrical parameters are listed in Table 2.

## 5. Experimental Results and Discussion

To provide insight into the properties and working principles of the proposed antenna, with and without the metamaterial, the simulated current distribution is analyzed and discussed. Figure 9 shows the current distribution at each operating frequency. The current path at the lower frequency 0.9 GHz is shown in Figure 9a with a maximum surface current observed strongly around the edges of the slots and a small portion of radiating elements as well as along the T- junction. Figure 9b,c shows the current distribution at 2.4 and 3.5 GHz. The intensity is high around the radiating elements and partially so on the slot edges. The radiation at 5.5 GHz is mainly from the radiating elements and the edge of the slots as shown in Figure 9d. Furthermore, the power divider also contributes almost at all frequencies.

As described in the literature, slots can be used to create a multiband operation [39]. By removing a small width and length of metal area, the antenna operating characteristics will be determined. As a result, the overall antenna performance will be enhanced. Here, it can be seen that multiband operation can be achieved due to the use of slots which strongly perturb the current distribution and increase the current path length as shown in Figure 9.

The impact of the rectangular slot (SL) is depicted in Figure 10a with a small impedance match at 3.5 GHz when the SL = 20 mm. Furthermore, the optimal benchmark SL value is 25 mm, while two additional resonances are generated at 0.9 and 2.4 GHz as shown in Figure 10b, whereas the two layers of the MTM unit cells are still applied.

Figure 11 shows the fabricated two-element antenna array prototype with the metamaterial unit cell. Its reflection coefficient (S11) with and without the MTM is shown in Figure 12. The S11 of the final measured prototype is seen in Figure 13, along with simulated results for the same antenna. It covers the four bands used for the following standards: GSM 900 at 900 MHz, WLAN at 2.3, 2.4, 5, 5.2, and 5.8 GHz, LTE-A at 3.5 GHz, and 5G WiFi centered at 5.5 GHz. The measured results indicate a significant increase in the bandwidth for the highest band, allowing improvement from 9.35% (simulated without unit cell) to 28% (measured with MTM). Moreover, the third band shifted slightly towards high frequencies at 3.5 GHz, with an increment of 27.81%. The lower band frequency at 900 MHz was matched by applying the MTM unit cell.

The achieved gain of the proposed antenna is shown in Figure 14. The maximum realized gain is increased from 3.4 (without metamaterial) to 6.47 dBi (with metamaterial) at 3.5 GHz. This is a 3 dB improvement and is due to the MTM layers acting as a superstrate which, unlike reflectors, use solid metal that can be placed in close proximity to the radiating elements (here 10 mm). Likewise, the obtained efficiency of the proposed antenna array increases significantly in the desired frequency bands by 35.7% at 3.5 GHz and 60% at 5.5 GHz as shown in Figure 15. The gain, bandwidth, and the overall size of the multiband antenna are summarized in Table 3 for each operational band.

To verify the performance of the proposed antenna loaded with metamaterial, Horn antenna was considered as transmitter source antenna, whereas the fabricated prototype was used as a receiver prior to being connected to an Agilent E8051C network analyzer (ENA) via a coaxial cable to measure the radiation patterns through a far-field measurement system which is located in an anechoic chamber. Both antennas terminals are placed 1.25 m apart. The E-field radiation patterns (XZ-plane and YZ-plane) shown in Figure 16 indicate almost an omnidirectional behavior at each resonant frequency except at 5.8 GHz, which indicates bidirectionality. Furthermore, the simulated and measured radiation patterns are in agreement.

## 6. Conclusions

A quad-band antenna array using a novel near-zero refractive index and double-negative metamaterial is presented. A compact SSRR resonator metamaterial unit cell was proposed and validated numerically, whereas NZRI and DNG properties exhibited negative permittivity, permeability, and a refractive index over several bands for all x and y principal axis wave propagation. The investigated MTM planes were then placed in two scenarios at the reverse and front sides of the proposed CPW antenna. The experimental results show that the proposed antenna integrated with the compact-sized MTM structure, which has proper impedance bandwidth covering the required bandwidths of GSM900, WLAN, LET-A, and 5G Wi-Fi in the 0.865–1.06, 2.24–2.52, 3.25–4.31, and 4.9–6.5 GHz, and stably omnidirectional E plane radiation patterns.

## Figures and Tables

**Figure 1 sensors-20-00457-f001:**
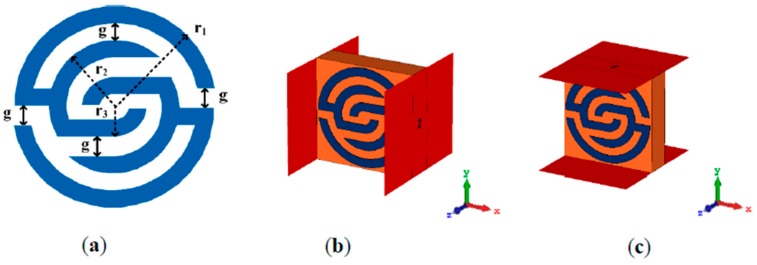
Metamaterial unit cell: (**a**) unit cell structure with g = 0.5 mm, r_1_ = 2.9 mm, r_2_ = 1.9 mm, r_3_ = 0.9 mm; (**b**) simulation set-up in the x-axis; (**c**) simulation set-up in the y-axis.

**Figure 2 sensors-20-00457-f002:**
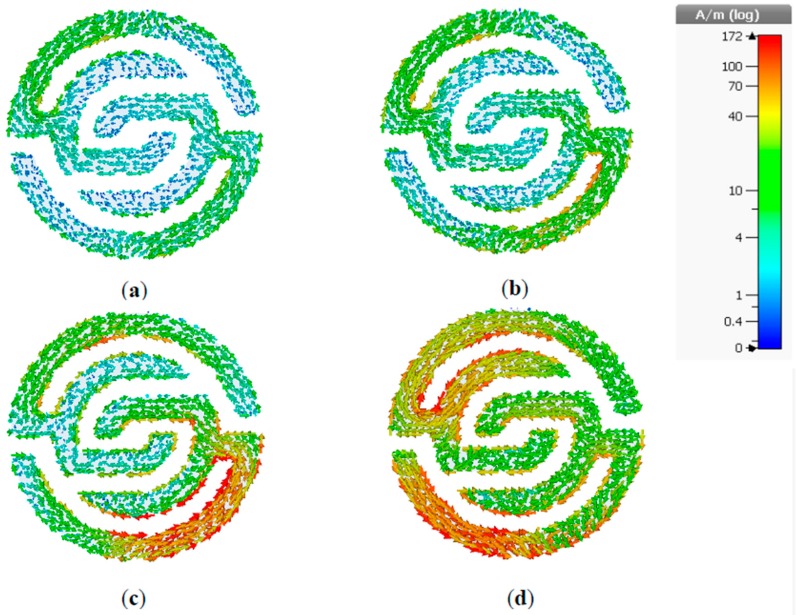
Surface current distribution at (**a**) 2.4; (**b**) 3.5; (**c**) 5.5 and (**d**) 10 GHz.

**Figure 3 sensors-20-00457-f003:**
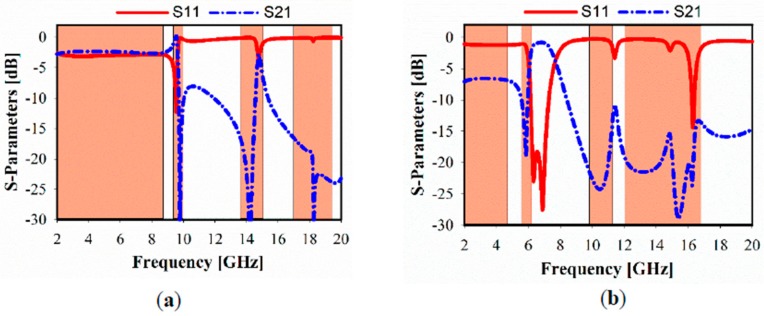
Simulated metamaterial reflection and transmission coefficients: (**a**) x-axis, (**b**) y-axis.

**Figure 4 sensors-20-00457-f004:**
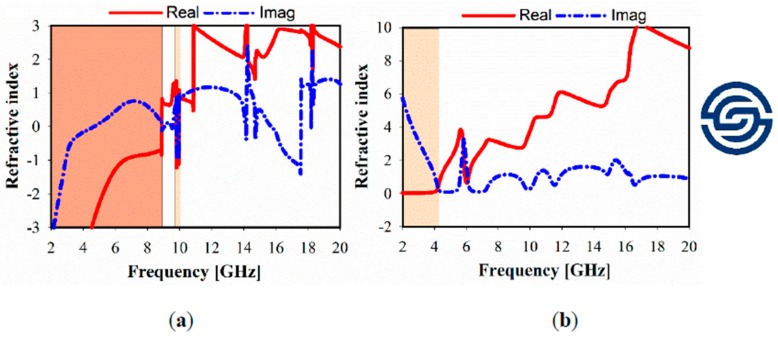
Metamaterial simulated refractive index of 1 × 1 unit cell: (**a**) x-axis; (**b**) y-axis.

**Figure 5 sensors-20-00457-f005:**
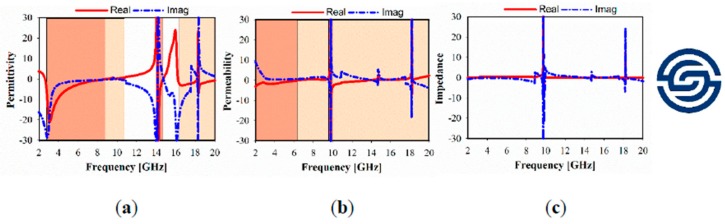
Metamaterial simulated results at x-axis: (**a**) permittivity; (**b**) permeability; and (**c**) impedance.

**Figure 6 sensors-20-00457-f006:**
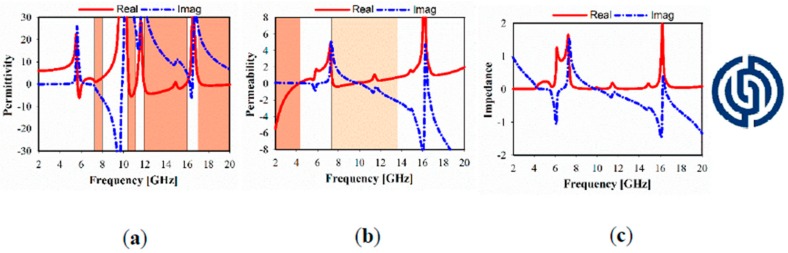
Metamaterial simulated result at y-axis: (**a**) permittivity; (**b**) permeability and (**c**) impedance.

**Figure 7 sensors-20-00457-f007:**
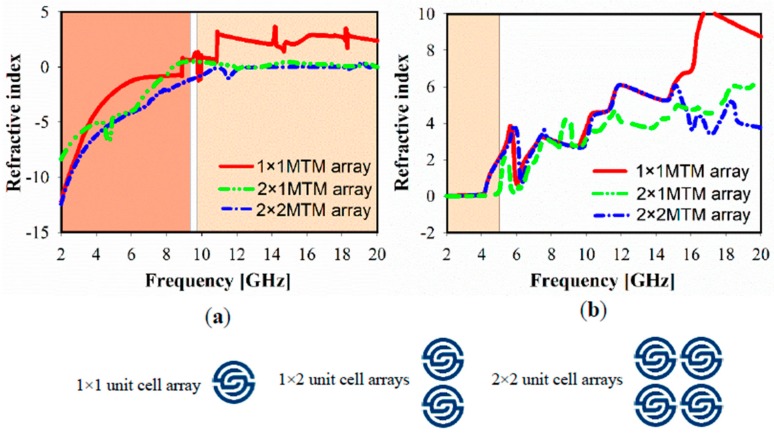
Metamaterial simulated refractive index results of the 1 × 1, 1 × 2 and 2 × 2 array structures: (**a**) x- axis; (**b**) y- axis.

**Figure 8 sensors-20-00457-f008:**
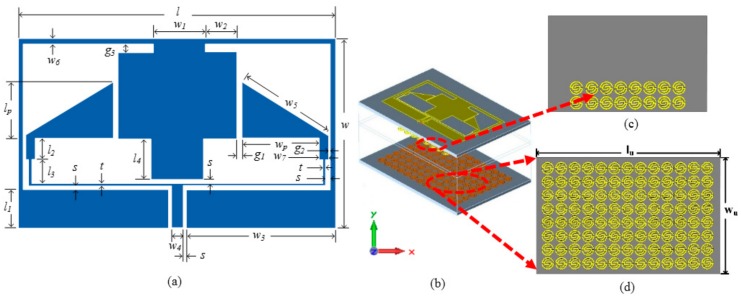
Geometry of the proposed antenna: (**a**) front view, (**b**) 3D view, (**c**) back view, (**d**) suspended separator metamaterial (MTM) layer.

**Figure 9 sensors-20-00457-f009:**
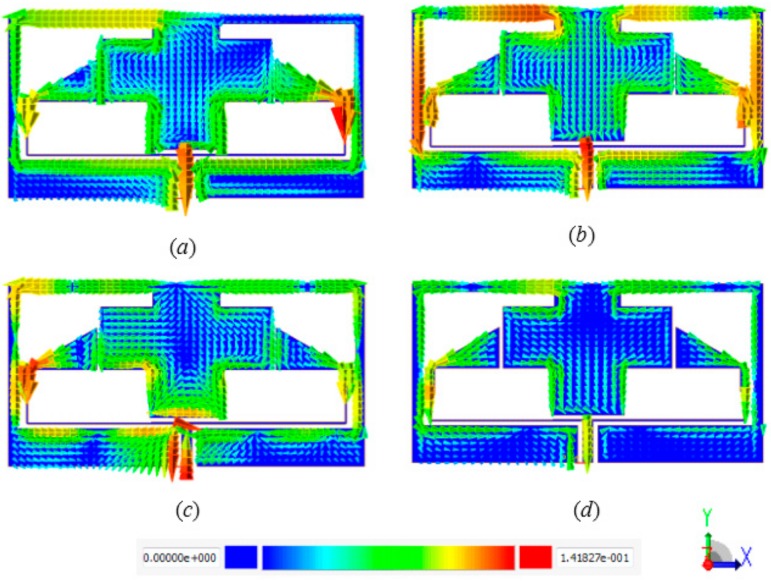
Simulated surface current intensity at: (**a**) 900; (**b**) 2.4; (**c**) 3.5; and (**d**) 5.5 GHz.

**Figure 10 sensors-20-00457-f010:**
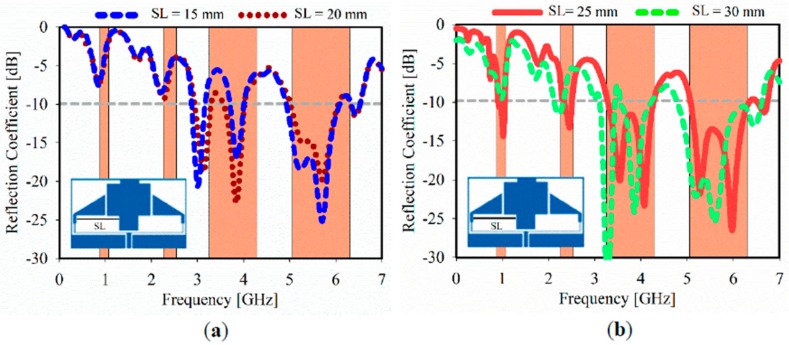
Simulated reflection coefficient for different slot length at: (**a**) SL = 15 mm, SL = 20 mm; (**b**) SL = 25 mm, and SL = 30 mm.

**Figure 11 sensors-20-00457-f011:**
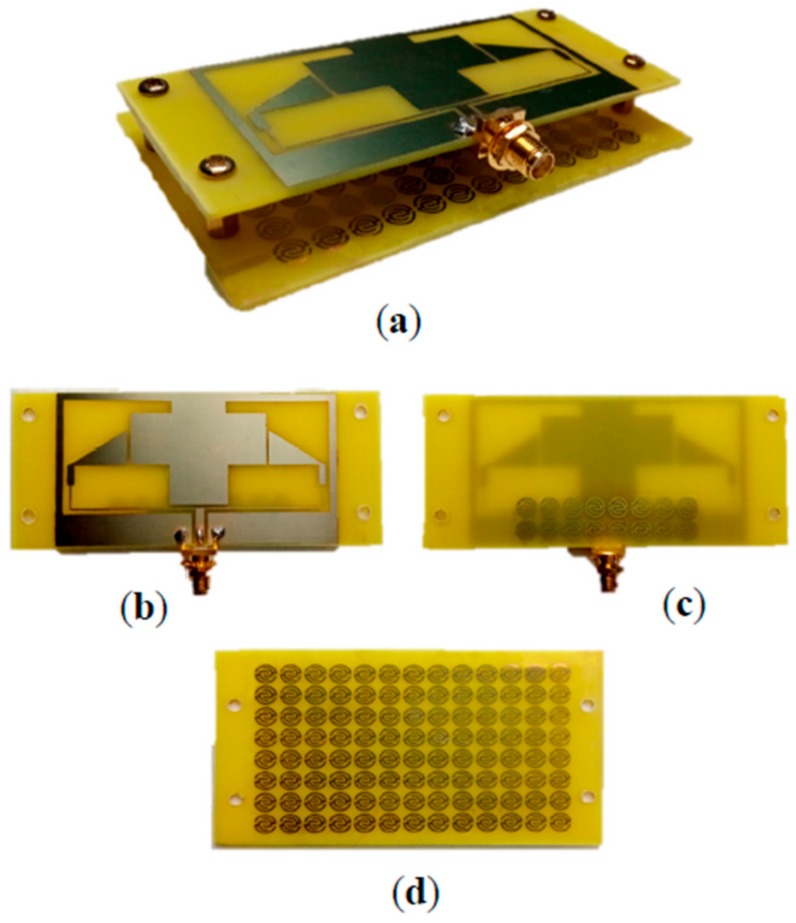
The fabricated prototype of the proposed coplanar waveguide (CPW) antenna: (**a**) 3D view; (**b**) front view; (**c**) back view with MTM; and (**d**) MTM super substrate structure.

**Figure 12 sensors-20-00457-f012:**
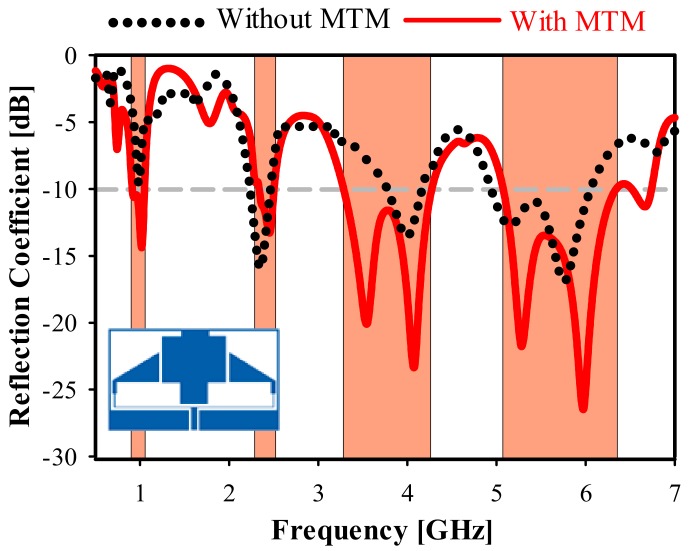
Simulated reflection coefficient (S11) with and without MTM.

**Figure 13 sensors-20-00457-f013:**
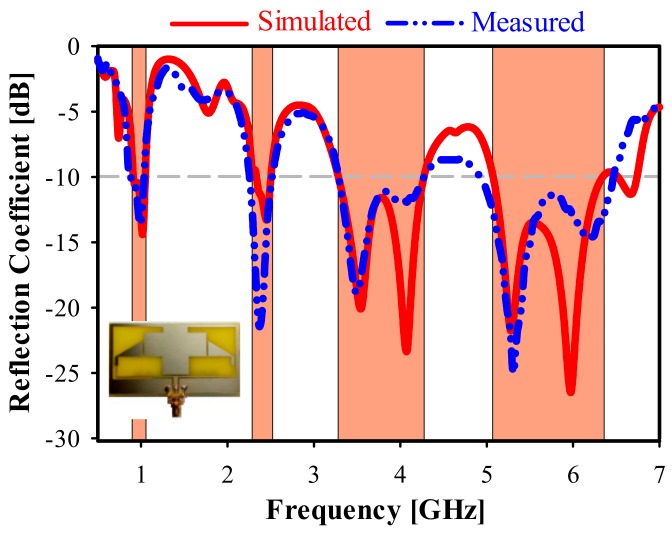
Simulated and measured reflection coefficient (S11) with MTM.

**Figure 14 sensors-20-00457-f014:**
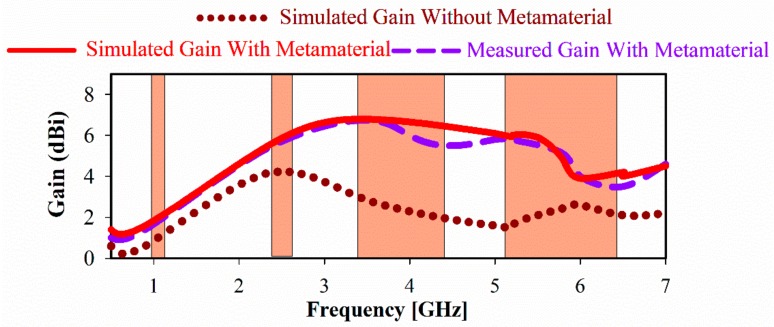
A measured and simulated gain of proposed CPW antenna with and without MTM.

**Figure 15 sensors-20-00457-f015:**
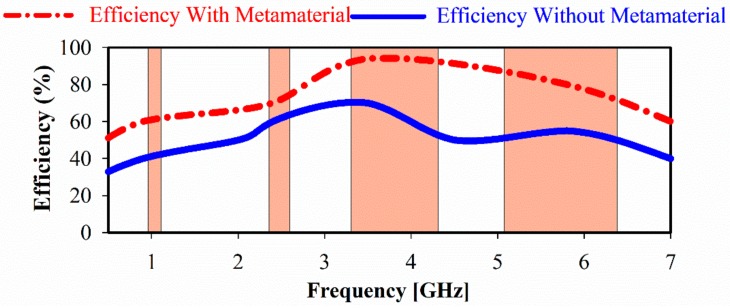
The proposed CPW antenna efficiency with and without MTM.

**Figure 16 sensors-20-00457-f016:**
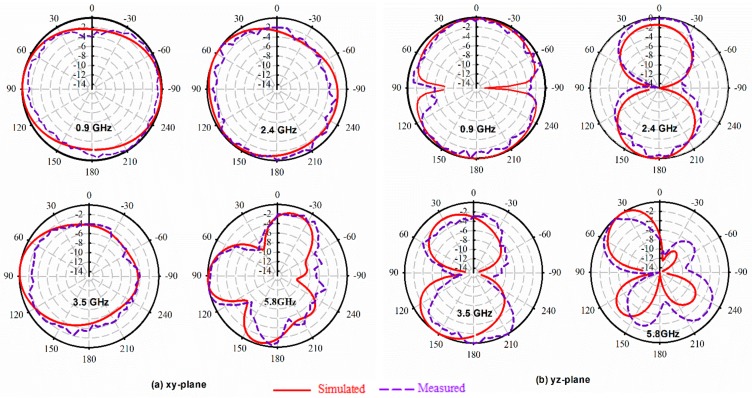
Simulated and measured electric field radiation patterns of the proposed CPW antenna at: the (**a**) XY-Plane; (**b**) YZ-Plane.

**Table 1 sensors-20-00457-t001:** Comparison of a designed antenna with others in the state of the art.

Reference	Size (mm^2^)	Operating Bands (GHz)	Technique	Max. BW (%)	Max. Gain(dBi)
[30]	45.0 × 45.5	(0.898–0.929), (1.540–1.580), (1.956–2.005)	CPW-Fed	02.47	2.72
[31]	30.0 × 30.0	(0.918–0.927), (2.440–2.460), (3.520–3.620), (5.700–6.270)	U, L, F-shaped	09.50	5.37
[32]	18.0 × 35.0	(1.660–2.710), (2.950–4.540), (5.020–6.100)	CPW-Fed	--	5.27
[33]	30.0 × 40.0	(2.400–2.700), (3.320–4.000), (4.760–5.800)	CPW-Fed & Slots	19.70	--
[34]	18.0 × 30.0	(1.765–2.695), (3.010–3.910), (5.110–6.055)	CPW-Fed	16.90	4.75
[35]	50.0 × 35.0	(1.680–2.040), (3.030–4.100), (4.760–6.840), (7.620–8.420)	I-shaped strips	35.80	-3.50
[36]	50.0 × 69.0	(1.430–1.600), (1.940–2.100), (2.400–2.570), (3.450–3.600)	L-shaped stub	06.80	1.38
[37]	85.0 × 125	(0.870–1.010), (1.720–1.960), (2.280–2.830), (5.710–6.380)	L-shaped radiator	11.10	5.82
[38]	40.0 × 20.0	(1.540–1.610), (2.310–2.720), (3.100–3.750), (5.030–5.950)	Asymmetric coplanar strip	16.70	3.50
[25]	24.8 × 30.0	(2.350–2.450), (2.630–2.760), (4.440–4.920), (5.420–5.770), (8.680–8.890)	Metamaterial	10.25	3.01
[26]	32.0 × 38.0	(2.400–2.600), (2.900–3.100), (3.300–3.500), (4.000–8.300)	Metamaterial	11.11	3.80
[27]	48.0 × 48.0	(1.710–1.880), (1.880–2.200), (3.400–3.800)	Metamaterial	13.00	4.72
[28]	125 × 125	(1.380–1.395), (1.570–1.580)	Artificial magnetic conductor (AMC)	02.00	7.00
This work	78.6 × 42.5	(0.865–1.060), (2.240–2.520), (3.250–4.310), (4.900–6.500)	NZRI & DNG Metamaterial	28.00	6.72

**Table 2 sensors-20-00457-t002:** Parameter dimensions of the proposed antenna.

Para.	Value (mm)	Para.	Value (mm)	Para.	Value (mm)	Para.	Value (mm)	Para.	Value (mm)
*l*	78.6	*g_1_*	0.766	*l_2_*	5.53	*w_3_*	37.1	*t*	0.42
*w*	42.5	*g_2_*	1.108	*l_3_*	6.506	*w_4_*	2.44	*s*	1
*l_p_*	9.87	*g_3_*	3.4	*l_4_*	11	*w_5_*	19.14		
*w_p_*	14.4	*l_1_*	8.86	*w_1_*	15	*w_6_*	2		
*w_u_*	55	*l_u_*	88	*w_2_*	11.5	*w_7_*	2		

**Table 3 sensors-20-00457-t003:** Summary of the antenna performance.

**Frequency Band (GHz)**	0.9	2.4	3.5	5.5
**Gain (dBi)**	1.89	5.05	6.74	5.98
**Bandwidth (%)**	20.26	11.76	27.81	28
**Efficiency (%)**	62	71	95	83
**Size (mm)**	78.6 × 42.5 × 0.035 m^3^			

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
