# Peer review of "Metamaterial Cell-Based Superstrate towards Bandwidth and Gain Enhancement of Quad-Band CPW-Fed Antenna for Wireless Applications"

_sensors, 2020, doi:10.3390/s20020457_

Round 1

Reviewer 1 Report

Section 3 is titled "Metamaterial unit cell working principle". It does include results for arrays as well and should be renamed accordingly. There are also a few incorrect figure references in Section 3, for example "Figure 2(a) to 4(d)" on Page 4, which I think should read "Figure 2(a) to 2(d)". Also on Page 4, it appears to me that the surface current is already somewhat visible in Figure 2(a), but intensifies in Figure 2(b), rather than being visible only in Figure 2(b). In general, more information in the figure captions would greatly improve the readability of this paper. In Figure 10, how is the variation of SL accomplished? Two drawings with different SL would be helpful.

Reviewer 2 Report

The paper discusses a novel type of antenna where a metamaterial is used as superstrate in order to enhance the gain and bandwidth of the antenna. Besides, the antenna allows to cover 4 bands of interest for communications at the same time.

The results are presented quite clearly, in particular it is appreciated the detailed analysis of the current distributions for the different cases and the comparison with and without metamaterial.

The grammar as well as the form in the paper should be considerably improved.

Concerning table 1 it is not clear why other implementations of antennas utilizing metamaterials are not considered as the authors do mention that there have been other attempts of joining metamaterials and antennas. It would be useful to write 1 or 2 sentences to make this point clear, which could also go to the advantage of the authors.

Given the dimensions of the antenna compared to some in Table 1 (e.g. ref 26), could the authors provide some perspectives on how to shrink the antenna dimensions?

Could the authors provide a few sentences about the experimental setup to measure the radiation performance?

Round 2

Reviewer 2 Report

I would like to thank the authors for having provided an improved version addressing the concerns that were raised. I think the manuscript in its current form is suitable for publication.